# How to compare adversarial robustness of classifiers from a global perspective

## Abstract

Adversarial robustness of machine learning models has attracted considerable attention over recent years. Adversarial attacks undermine the reliability of and trust in machine learning models, but the construction of more robust models hinges on a rigorous understanding of adversarial robustness as a property of a given model. Point-wise measures for specific threat models are currently the most popular tool for comparing the robustness of classifiers and are used in most recent publications on adversarial robustness. In this work, we use robustness curves to show that point-wise measures fail to capture important global properties that are essential to reliably compare the robustness of different classifiers. We introduce new ways in which robustness curves can be used to systematically uncover these properties and provide concrete recommendations for researchers and practitioners when assessing and comparing the robustness of trained models. Furthermore, we characterize scale as a way to distinguish small and large perturbations, and relate it to inherent properties of data sets, demonstrating that robustness thresholds must be chosen accordingly. We hope that our work contributes to a shift of focus away from point-wise measures of robustness and towards a discussion of the question what kind of robustness could and should reasonably be expected. We release code to reproduce all experiments presented in this paper, which includes a Python module to calculate robustness curves for arbitrary data sets and classifiers, supporting a number of frameworks, including TensorFlow, PyTorch and JAX.

## 1 Introduction

Despite their astonishing success in a wide range of classification tasks, deep neural networks can be lead to incorrectly classify inputs altered with specially crafted adversarial perturbations (Szegedy et al. 2014; Goodfellow et al. 2015). These perturbations can be so small that they remain almost imperceptible to human observers (J. P. Göpfert et al. 2020). Adversarial robustness describes a model's ability to behave correctly under such small perturbations crafted with the intent to mislead the model. The study of adversarial robustness – with its definitions, their implications, attacks, and defenses – has attracted considerable research interest. This is due to both the practical importance of trustworthy models as well as the intellectual interest in the differences between decisions of machine learning models and our human perception. A crucial starting point for any such analysis is the definition of what exactly a small input perturbation is – requiring (a) the choice of a *distance function* to measure perturbation size, and (b) the choice of a particular *scale* to distinguish small and large perturbations. Together, these two choices determine a *threat model* that defines exactly under which perturbations a model is required to be robust.

The most popular choice of distance function is the class of distances induced by $\ell_p$ norms (Szegedy et al. 2014; Goodfellow et al. 2015; Carlini, Athalye, et al. 2019), in particular $\ell_1, \ell_2$ and $\ell_\infty$, although other choices such as Wasserstein distance have been explored as well (Wong, Schmidt, et al. 2019). Regarding scale, the current default is to pick some perturbation threshold $\varepsilon$ without providing concrete reasons for the exact choice. Analysis then focuses on the *robust error* of the model, the proportion of test inputs for which the model behaves incorrectly under some perturbation up to size $\varepsilon$. This means that the scale is defined as a binary distinction between small and large perturbations based on the perturbation threshold. A set of canonical thresholds have emerged in

the literature. For example, in the publications referenced in this section, the `MNIST` data set is typically evaluated at a perturbation threshold $\varepsilon \in \{0.1, 0.3\}$ for the $\ell_\infty$ norm, while `CIFAR-10` is evaluated at $\varepsilon \in \{2/255, 4/255, 8/255\}$, stemming from the three 8-bit color channels used to represent images.

Based on these established threat models, researchers have developed specialized methods to minimize the robust error during training, which results in more robust models. Popular approaches include specific data augmentation, sometimes used under the umbrella term adversarial training (Guo et al. 2017; Madry et al. 2018; Carmon et al. 2019; Hendrycks et al. 2019), training under regularization that encourages large margins and smooth decision boundaries in the learned model (Hein and Andriushchenko 2017; Wong and Kolter 2018; Croce, Andriushchenko, and Hein 2019; Croce and Hein 2020), and post-hoc processing or randomized smoothing of predictions in a learned model (Lecuyer et al. 2019; Cohen et al. 2019).

In order to show the superiority of a new method, robust accuracies of differently trained models are typically compared for a handful of threat models and data sets, eg., $\ell_\infty(\varepsilon = 0.1)$ and $\ell_2(\varepsilon = 0.3)$ for `MNIST`. Out of 22 publications on adversarial robustness published at NeurIPS 2019, ICLR 2020, and ICML 2020, 12 publications contain results for only a single perturbation threshold. In five publications, robust errors are calculated for at least two different perturbation thresholds, but still, only an arbitrary number of thresholds is considered. Only in five out of the total 22 publications do we find extensive considerations of different perturbation thresholds and the respective robust errors. Out of these five, three are analyses of randomized smoothing, which naturally gives rise to certification radii (B. Li et al. 2019; Carmon et al. 2019; Pinot et al. 2019). Najafi et al. (2019) follow a learning-theoretical motivation, which results in an error bound as a function of the perturbation threshold. Only Maini et al. (2020) do not rely on randomization and still provide a complete, empirical analysis of robust error for varying perturbation thresholds[1].

*Our contributions:* In this work, we demonstrate that point-wise measures of $\ell_p$ robustness are not sufficient to reliably and meaningfully compare the robustness of different classifiers. We show that, both in theory and practice, results of model comparisons based on point-wise measures may fail to generalize to threat models with even slightly larger or smaller $\varepsilon$ and that robustness curves avoid this pitfall by design. Furthermore, we show that point-wise measures are insufficient to meaningfully compare the efficacy of different defense techniques when distance functions are varied, and that robustness curves, again, are able to reliably detect and visualize this property. Finally, we analyze how scale depends on the underlying data space, choice of distance function, and distribution. Based on our findings we suggest that robustness curves should become the standard tool when comparing adversarial robustness of classifiers, and that the perturbation threshold of threat models should be selected carefully in order to be meaningful, considering inherent characteristics of the data set. We release code to reproduce all experiments presented in this paper[2], which includes a Python module with an easily accessible interface (similar to Foolbox, Rauber et al. (2017)) to calculate robustness curves for arbitrary data sets and classifiers. The module supports classifiers written in most of the popular machine learning frameworks, such as TensorFlow, PyTorch and JAX.

## 2 METHODS

An adversarial perturbation for a classifier $f$ and input-output pair $(x, y)$ is a small perturbation $\delta$ with $f(x + \delta) \neq y$. Because the perturbation $\delta$ is small, it is assumed that the label $y$ would still be the correct prediction for $x + \delta$. The resulting point $x + \delta$ is called an adversarial example. The points vulnerable to adversarial perturbations are the points that are either already misclassified when unperturbed, or those that lie close to a decision boundary.

One tool to visualize and study the robustness behavior of a classifier are *robustness curves*, first used by Wong and Kolter (2018) and later formalized by C. Göpfert et al. (2020). A robustness curve

---

[1]*Single thresholds:* (Mao et al. 2019; Tramer and Boneh 2019; Alayrac et al. 2019; Brendel et al. 2019; Qin et al. 2019; Wang et al. 2020; Song et al. 2020; Croce and Hein 2020; Xie and Yuille 2020; Rice et al. 2020; Zhang et al. 2020; Singla and Feizi 2020), *multiple thresholds:* (Lee et al. 2019; Mahloujifar et al. 2019; Hendrycks et al. 2019; Wong, Rice, et al. 2020; Boopathy et al. 2020), *full analysis:* (Pinot et al. 2019; Carmon et al. 2019; B. Li et al. 2019; Najafi et al. 2019; Maini et al. 2020).

[2]The full code is available at https://github.com/Anonymous23984902384/how-to-compare-adversarial-robustness-of-classifiers-from-a-global-perspective.

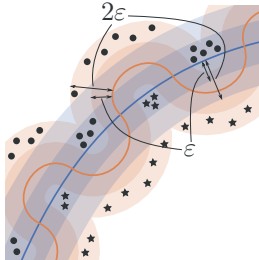 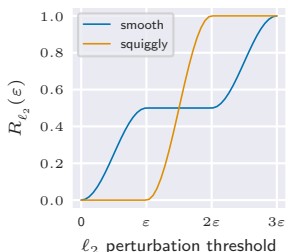

Figure 1: Excerpt of a toy data set with two decision boundaries (left) and respective robustness curves (right). The data is separated perfectly by one smooth boundary (blue robustness curve), and one squiggly boundary (orange robustness curve). We indicate margins around the boundaries at distances $\varepsilon$ and $2\varepsilon$. Selecting a single perturbation threshold is not sufficient to decide which classifier is more robust.

captures the distribution of shortest distances between a set of points and the decision boundaries of a classifier:

**Definition 1.** *Given an input space $\mathcal{X}$ and label set $\mathcal{Y}$, distance function $d$ on $\mathcal{X} \times \mathcal{X}$, and classifier $f : \mathcal{X} \to \mathcal{Y}$. Assume $(x, y) \sim_{i.i.d.} P$ for some distribution $P$ on $\mathcal{X} \times \mathcal{Y}$. Then the $d$-robustness curve for $f$ is the graph of the function*

$$R_d^f(\varepsilon) := P\left(\{(x, y) \ s.t. \ \exists \ x' : d(x, x') \leqslant \varepsilon \wedge f(x') \neq y\}\right)$$

A model's robustness curve shows how data points are distributed in relation to the decision boundaries of the model, essentially visualizing simultaneously an extremely large number of point-wise measures. This allows us to take a step back from robustness regarding a specific perturbation threshold and instead compare global robustness for different classifiers, distributions and distance functions. To see why this is relevant, consider Figure 1, which shows toy data along with two possible classifiers that perfectly separate the data. For a perturbation threshold of $\varepsilon$, the blue classifier has robust error $0.5$, while the orange classifier is perfectly robust. However, for a perturbation threshold of $2\varepsilon$, the orange classifier has robust error $1$, while the blue classifier remains at $0.5$. By freely choosing a single perturbation threshold for comparison, it is therefore possible to make either classifier appear to be much better than the other, and no single threshold can capture the whole picture. In fact, for any two disjoint sets of perturbation thresholds, it is possible to construct a data distribution and two classifiers $f$, $f'$, such that the robust error of $f$ is lower than that of $f'$ for all perturbation thresholds in the first set, and that of $f'$ is lower than that of $f$ for all perturbation thresholds in the second set. See Appendix A for a constructive proof. This shows that even computing multiple point-wise measures to compare two models may give misleading results.

## 3   EXPERIMENTS

In the following, we empirically evaluate the robustness of a number of recently published models, and demonstrate that the weaknesses of point-wise measures described above are not limited to toy examples, but occur for real-world data and models.

### 3.1   EXPERIMENTAL SETUP

We evaluate and compare the robustness of models obtained using the following training methods:

1. Standard training (`ST`), i. e., training without specific robustness considerations.
2. Adversarial training (`AT`) (Madry et al. 2018).
3. Training with robust loss (`KW`) (Wong and Kolter 2018).
4. Maximum margin regularization for a single $\ell_p$ norm together with adversarial training (`MMR+AT`) (Croce, Andriushchenko, and Hein 2019).
5. Maximum margin regularization simultaneously for $\ell_\infty$ and $\ell_1$ margins (`MMR-UNIV`) (Croce and Hein 2020).

Table 1: Three point-wise measures for different threat models. All threat models use the $\ell_\infty$ distance function, but differ in choice of perturbation threshold (denoted by $\varepsilon$). Each row contains the robust test errors for one point-wise measure. Each column contains the robust test errors for one model, trained with a specific training method (marked by column title). The lower the number, the better the robustness for the specific threat model. Each point-wise measure results in a different relative ordering of the classifiers based on the errors. The order is visualized by different tones of gray in the background of the cells.

| $\varepsilon$ | ST | AT | KW | MMR + AT | MMR-UNIV |
|---|---|---|---|---|---|
| 1/255 | 0.60 | 0.38 | 0.43 | 0.42 | 0.54 |
| 4/255 | 0.99 | 0.68 | 0.57 | 0.63 | 0.74 |
| 8/255 | 1.00 | 0.92 | 0.73 | 0.84 | 0.91 |

Together with each training method, we state the threat model the trained model is optimized to defend against, eg., $\ell_\infty(\varepsilon = 0.1)$ for perturbations in $\ell_\infty$ norm with perturbation threshold $\varepsilon = 0.1$, if any. The trained models are those made publicly available by Croce, Andriushchenko, and Hein (2019)[3] and Croce and Hein (2020)[4]. The network architecture is a convolutional network with two convolutional layers, two fully connected layers and ReLU activation functions. The evaluation is based on six real-world datasets: MNIST, Fashion-MNIST (FMNIST) (Xiao et al. 2017), German Traffic Signs (GTS) (Houben et al. 2013), CIFAR-10 (Krizhevsky 2009), Tiny-Imagenet-200 (TINY-IMG) (F.-F. Li et al. 2016), and Human Activity Recognition (HAR) (Anguita et al. 2013). For specifics on model training (hyperparameters, architecture details), refer to Appendix C. Models are generally trained on the full training set for the corresponding data set, and robustness curves evaluated on the full test set, unless stated otherwise.

For complex models, calculating the exact distance of a point to the closest decision boundary, and thus estimating the true robustness curve, is computationally very intensive, if not intractable. Therefore we bound the true robustness curve from below using strong adversarial attacks, which is consistent with the literature on empirical evaluation of adversarial robustness and also applicable to many different types of classifiers. We base our selection of attacks on the recommendations by Carlini, Athalye, et al. (2019). Specifically, we use the $\ell_2$-attack proposed by (Carlini and Wagner 2017) for $\ell_2$ robustness curves and PGD (Madry et al. 2018) for $\ell_\infty$ robustness curves. For both attacks, we use the implementations of Foolbox (Rauber et al. 2017). See Appendix C for information on adversarial attack hyperparameters. In the following, "robustness curve" refers to this empirical approximation of the true robustness curve.

## 3.2 THE WEAKNESSES OF POINT-WISE MEASURES

Point-wise measures are used to quantify robustness of classifiers by measuring the robust test error for a specific distance function and a perturbation threshold (eg., $\ell_\infty(\varepsilon = 4/255)$). In Table 1 we show three point-wise measures to compare the robustness of five different classifiers on CIFAR-10. If we compare the robustness of the four robust training methods (latter four columns of the table) based on the first point-wise threat model $\ell_\infty(\varepsilon = 1/255)$ (first row of the table), we can see that the classifier trained with AT is the most robust, followed by MMR + AT, followed by KW, and MMR-UNIV results in the least robust classifier. However, if we increase the $\varepsilon$ of our threat model to $\varepsilon = 4/255$ (second row of the table), KW is more robust than AT. For a even larger $\varepsilon$ (third row of the table), we would conclude that MMR-UNIV is preferable over AT, and that AT results in the least robust classifier. All three statements are true for the particular perturbation threshold ($\varepsilon$), and the magnitude of all perturbation thresholds is reasonable: publications on adversarial robustness typically evaluate CIFAR-10 on perturbation thresholds $\leqslant 10/255$ for $\ell_\infty$ perturbations. Meaningful conclusions on the robustness of the classifiers relative to each other can not be made without taking all possible $\varepsilon$ into account. In other words, a global perspective is needed.

---

[3]The models trained with ST, KW, AT and MMR + AT are avaible at
www.github.com/max-andr/provable-robustness-max-linear-regions.
[4]The models trained with MMR-UNIV are avaible at www.github.com/fra31/mmr-universal.

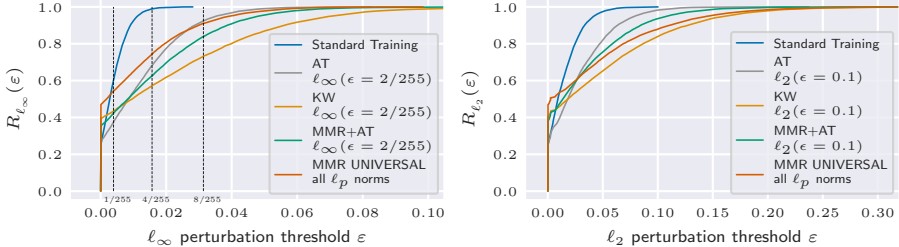

Figure 2: $\ell_\infty$ robustness curves (left plot) and $\ell_2$ robustness curves (right plot) resulting from different training methods (indicated by label), optimized for different threat models (indicated by label). The dashed vertical lines visualize the three point-wise measures from Table 1. The models are trained and evaluated on the full training-/test sets of CIFAR-10. The curves allow us to reliably compare the robustness of the classifiers, unbiased by choice of perturbation threshold.

### 3.2.1 A GLOBAL PERSPECTIVE

Figure 2 shows the robustness of different classifiers for the $\ell_\infty$ (right plot) and $\ell_2$ (left plot) distance functions from a global perspective using robustness curves. The plot reveals why the three point-wise measures (marked by vertical black dashed lines in the left plot) lead to different results in the relative ranking of robustness of the classifiers. Both for the classifiers trained to be robust against attacks in $\ell_\infty$ distance (left plot) and $\ell_2$ distance (right plot), we can observe multiple intersections of robustness curves, corresponding to changes in the relative ranking of the robustness of the compared classifiers. The robustness curves allow us to reliably compare the robustness of classifiers for all possible perturbation thresholds. Furthermore, the curves clearly show the perturbation threshold intervals with strong and weak robustness for each classifier, and are not biased by an arbitrarily chosen perturbation threshold.

### 3.2.2 OVERFITTING TO SPECIFIC PERTURBATION THRESHOLDS

In addition to the problem of robustness curve intersection, relying on point-wise robustness measures to evaluate adversarial robustness is prone to overfitting when designing training procedures. Figure 3 shows $\ell_\infty$ robustness curves for MMR + AT with $\ell_\infty$ threat model as provided by Croce, Andriushchenko, and Hein (2019). The models trained on MNIST and FMNIST both show a change in slope, which could be a sign of overfitting to the specific threat models for which the classifiers were optimized for, since the change of slope occurs approximately at the chosen perturbation threshold $\varepsilon$. This showcases a potential problem with the use of point-wise measures during training. The binary separation of "small" and "large" perturbations based on the perturbation threshold is not sufficient to capture the intricacies of human perception under perturbations, but a simplification based on the idea that perturbations below the perturbation threshold should almost certainly not lead to a change in classification. If a training procedure moves decision boundaries so that data points lie just beyond this threshold, it may achieve a low robust error, without furthering the actual goals of adversarial robustness research. Using robustness curves for evaluation cannot prevent this effect, but can be used to detect it.

### 3.2.3 TRANSFER OF ROBUSTNESS ACROSS DISTANCE FUNCTIONS

In the following, we analyze to which extent properties of robustness curves transfer across different choices of distance functions. If properties transfer, it may not be necessary to individually analyze robustness for each distance function.

In Figure 4 we compare the robustness of different models for the $\ell_\infty$ (left plot) and $\ell_2$ (right plot) distance functions. The difference to Figure 2 is that the models (indicated by colour) are the same models in the left plot and in the right plot. We find that for MMR + AT, the $\ell_\infty$ threat model leads to better robustness than the $\ell_2$ threat model *both* for $\ell_\infty$ *and* $\ell_2$ robustness curves. In fact, MMR + AT with the $\ell_\infty$ threat model even leads to better $\ell_\infty$ and $\ell_2$ robustness curves than MMR-UNIV, which is specifically designed to improve robustness for all $\ell_p$ norms. Overall, the plots are visually similar.

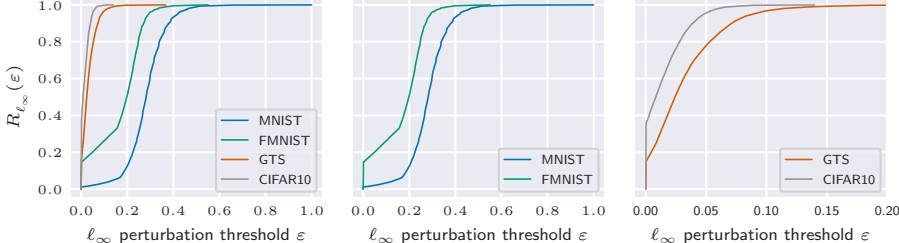

Figure 3: $\ell_\infty$ robustness curves for multiple data sets. Each curve is calculated for a different model and a different test data set. The data sets are indicated by the labels. The models are trained with MMR + AT, Threat Models: MNIST: $\ell_\infty(\varepsilon = 0.1)$, FMNIST: $\ell_\infty(\varepsilon = 0.1)$, GTS: $\ell_\infty(\varepsilon = 4/255)$, CIFAR-10: $\ell_\infty(\varepsilon = 2/255)$. The curves for MNIST and FMNIST both show a change in slope, which can not be captured with point-wise measures and could be a sign of overfitting to the specific threat models for which the classifiers were optimized for.

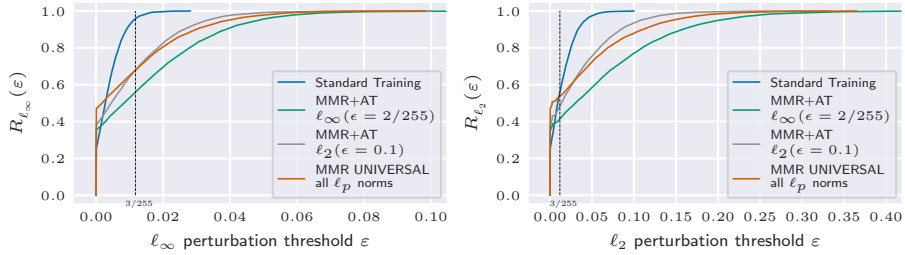

Figure 4: $\ell_\infty$ robustness curves (left plot) and $\ell_2$ robustness curves (right plot) resulting from different training methods (indicated by color and label), optimized for different threat models (indicated by label). The models are trained and evaluated on the full training-/test sets of CIFAR-10. The curves allow us to reliably compare the transfer of robustness of the classifiers across distance functions, unbiased by choice of threat model.

However, since both plots contain multiple robustness curve intersections, the ranking of methods remains sensitive to the choice of perturbation threshold. For example, a perturbation threshold of $\varepsilon = 3/255$ (vertical black dashed line) for the $\ell_\infty$ distance function (left subplot) shows that the classifier trained with MMR + AT ($\ell_2(\varepsilon = 0.1)$) is approximately as robust as the classifier trained with MMR-UNIV. The same perturbation threshold for the $\ell_2$ distance function (right subplot) shows that the classifier trained with MMR + AT is more robust than the classifier trained with MMR-UNIV for $\ell_2$ threat models. Using typical perturbation thresholds from the literature for each distance function does not alleviate this issue: At perturbation threshold $\varepsilon = 2/255$ for $\ell_\infty$ distance, the classifier trained with MMR + AT ($\ell_2(\varepsilon = 0.1)$) is more robust than the one trained with MMR-UNIV, while at perturbation threshold $\varepsilon = 0.1$ for $\ell_2$ distance, the opposite is true. This shows that even when robustness curves across various distance functions are qualitatively similar, this may be obscured by the choice of threat model(s) to compare on.

We also emphasize that in general, robustness curves across various distance functions may be qualitatively *dis*similar. In particular:

1. For linear classifiers, the *shape* of a robustness curve is identical for distances induced by different $\ell_p$ norms. This follows from Theorem 2 in Appendix B, which is an extension of a weaker result in C. Göpfert et al. (2020). For non-linear classifiers, different $\ell_p$ norms may induce different robustness curve shapes. See C. Göpfert et al. (2020) for an example.
2. Even for linear classifiers, robustness curve *intersections* do not transfer between distances induced by different $\ell_p$ norms. That is, for two linear classifiers, there may exist $p, p'$ such that the robustness curves for the $\ell_p$ distance intersect, but not the robustness curves for the $\ell_{p'}$ distance. See Appendix A for an example.

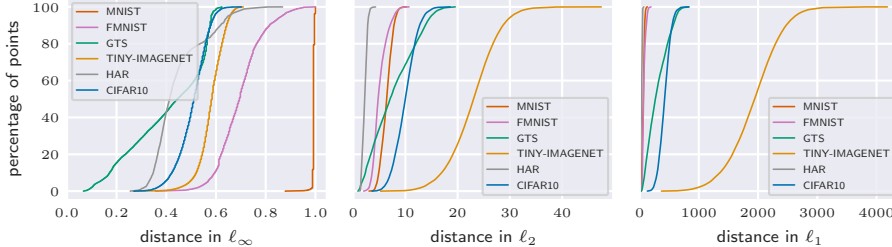

Figure 5: Minimum inter-class distances of all data sets considered in this work, measured in $\ell_\infty$ (left), $\ell_2$ (middle), and $\ell_1$ (right) norm. See Table 2 for size and dimensionality. The shapes of the curves and the threshold from which any classifier must necessarily trade of between accuracy and robustness differ strongly between data sets.

### 3.3   ON THE RELATIONSHIP BETWEEN SCALE AND DATA

As the previous sections show, robustness curves can be used to reveal properties of robust models that may be obscured by point-wise measures. However, some concept of scale, that is, some way to judge whether a perturbation is small or large, remains necessary. *Especially* when robustness curves intersect, it is crucial to be able to judge how critical it is for a model to be stable under the given perturbations. For many pairs of distance function and data set, canonical perturbation thresholds have emerged in the literature, but to the best of our knowledge, no reasons for these choices are given.

Since the assumption behind adversarial examples is that small perturbations should not affect classification behavior, the question of scale cannot be answered independently of the data distribution. In order to understand how to interpret different perturbation sizes, it can be helpful to understand how strongly the data point would need to be perturbed to *actually* change the *correct* classification. We call this the *inter-class distance* and analyze the distribution of inter-class distances for several popular data sets.

In Figure 5 we compare the inter-class distance distributions in $\ell_\infty$, $\ell_2$, and $\ell_1$ norm for all data sets considered in this work. We observe that for the $\ell_1$ and $\ell_2$ norms, the shape of the curves is similar across data sets, but their extent is determined by the dimensionality of the data space. In the $\ell_\infty$ norm, vastly different curves emerge for the different data sets. We hypothesize that, because the inter-class distance distributions vary more strongly for $\ell_\infty$ distances than for $\ell_1$ distances, the results of robustifying a model w. r. t. $\ell_\infty$ distances may depend more strongly on the underlying data distribution than the results of robustifying w. r. t. $\ell_1$ distances. This is an interesting avenue for future work.

When we look at the smallest inter-class distances in the $\ell_\infty$ norm (where all distances lie in the interval $[0, 1]$), we can make several observations. Because the smallest inter-class distance for MNIST in the $\ell_\infty$ norm is around $0.9$, we can see that transforming an input from one class to one from a different class almost always requires completely flipping at least one pixel from almost-black to almost-white or vice versa. For the other datasets, the inter-class distance distributions are more spread out than the inter-class distance distribution of MNIST. We observe that for CIFAR-10 with $\ell_\infty$ perturbations of size $\geqslant 0.25$, it becomes possible to transform samples from different classes into each other, so starting from this threshold, any classifier must necessarily trade off between accuracy and robustness. The shapes of the curves and the threshold from which any classifier must necessarily trade of between accuracy and robustness differ strongly between data sets – refer to Table 2 for exact values for the threshold.

In Table 2, we summarize the smallest and largest inter-class distances in different norms together with additional information about the size, number of classes, and dimensionality of the all the data sets we consider in this work. The values correspond directly to Figure 5, but even in this simplified view, we can quickly make out key differences between the data sets. Compare, for example, MNIST and GTS: While it appears reasonable to expect $\ell_\infty$ robustness of $0.3$ for MNIST, the same threshold for GTS is not possible. Relating Table 2 and Figure 3, we find entirely plausible the strong robustness

Table 2: Smallest and largest inter-class distances for subsets of several data sets, measured in $l_\infty$, $l_2$, and $l_1$ norm, together with basic contextual information about the data sets. All data has been been normalized to lie within the interval $[0, 1]$, and duplicates and corrupted data points have been removed. Apart from HAR, all data sets contain images – the dimensionality reported specifies their sizes and number of channels.

| | | | | Inter-class Distance | | | | | |
| | | | | Smallest | | | Largest | | |
| Dataset | Samples | Classes | Dimensionality | $l_\infty$ | $l_2$ | $l_1$ | $l_\infty$ | $l_2$ | $l_1$ |
|---|---|---|---|---|---|---|---|---|---|
| MNIST | 10 000 | 10 | $28 \times 28 \times 1$ | 0.88 | 3.03 | 19.16 | 1.00 | 10.18 | 132.38 |
| TINY-IMG | 98 139 | 200 | $64 \times 64 \times 3$ | 0.27 | 5.24 | 369.29 | 0.71 | 47.49 | 4184.37 |
| FMNIST | 10 000 | 10 | $28 \times 28 \times 1$ | 0.36 | 2.00 | 24.87 | 1.00 | 10.70 | 194.29 |
| GTS | 10 000 | 43 | $32 \times 32 \times 3$ | 0.07 | 0.90 | 31.46 | 0.62 | 19.54 | 833.22 |
| CIFAR-10 | 10 000 | 10 | $32 \times 32 \times 3$ | 0.27 | 3.61 | 130.77 | 0.70 | 18.57 | 831.44 |
| HAR | 2947 | 6 | 561 | 0.26 | 1.26 | 12.95 | 0.87 | 4.29 | 73.19 |

results for MNIST, and the small perturbation threshold for GTS. Based on inter-class distances we also expect less $\ell_\infty$ robustness for CIFAR-10 than for FMNIST, but not as seen in Figure 3. In any case, it is safe to say that, when judging the robustness of a model by a certain threshold, that number must be set with respect to the distribution the model operates on.

Overall, the strong dependence of robustness curves on the data set and the chosen norm, emphasizes the necessity of informed and conscious decisions regarding robustness thresholds. We provide an easily accessible reference in the form of Table 2, that should prove useful while judging scales in a threat model.

## 4 DISCUSSION

We have demonstrated that comparisons of robustness of different classifiers using point-wise measures can be heavily biased by the choice of perturbation threshold and distance function of the threat model, and that conclusions about rankings of classifiers with regards to their robustness based on point-wise measures therefore only provide a narrow view of the actual robustness behavior of the classifiers. Further, we have demonstrated different ways of using robustness curves to overcome the shortcomings of point-wise measures, and therefore recommend using them as the standard tool for comparing the robustness of classifiers. Finally, we have demonstrated how suitable perturbation thresholds necessarily depend on the data they pertain to.

It is our hope that practitioners and researchers alike will use the methodology proposed in this work, especially when developing and comparing adversarial defenses, and carefully motivate any concrete threat models they might choose, taking into account all available context.

*Limitations:* Computing approximate robustness curves for state-of-the-art classifiers and large data sets is computationally very intensive, due to the need of computing approximate minimal adversarial perturbations with strong adversarial attacks. Developing adversarial attacks which are both strong and fast is an ongoing challenge in the field of adversarial robustness.

One way to reduce the computational cost is to approximate the robustness curves by computing a set of point-wise measures. However, since robustness curves may intersect at arbitrarily many points, this may give misleading results. It would be interesting to investigate how closely robustness curves need to be approximated in order to estimate the number of intersections, if any, and their location, with high certainty.

Another limitation of our work is the focus on a small group of distance functions (mainly $\ell_\infty$ and $\ell_2$ norms). Even though it does intuitively make sense that models should at least be robust against these types of perturbations, a more general evaluation able to consider more distance functions simultaneously could be advantageous.

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

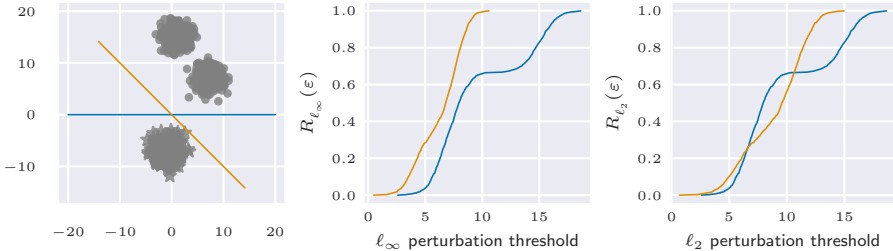

Figure 6: Example of a data distribution and two linear classifiers such that the $\ell_2$ robustness curves intersect, but not the $\ell_\infty$ robustness curves.

Cihang Xie and Alan Yuille (Apr. 2020). "Intriguing Properties of Adversarial Training at Scale". en. In.

Jingfeng Zhang, Xilie Xu, Bo Han, Gang Niu, Lizhen Cui, Masashi Sugiyama, and Mohan Kankanhalli (2020). "Attacks Which Do Not Kill Training Make Adversarial Learning Stronger". en. In: *Proceedings of the International Conference on Machine Learning* 1.

## A    ROBUSTNESS CURVES WITH ARBITRARY INTERSECTIONS

**Theorem 1.** *Let $T_1, T_2 \subset \mathbb{R}^{>0}$ be two disjoint finite sets. Then there exists a distribution $P$ on $\mathbb{R} \times \{0, 1\}$ and two classifiers $c_1, c_2 : \mathbb{R} \to \{0, 1\}$ such that $R^{c_1}_{|\cdot|}(t) < R^{c_2}_{|\cdot|}(t)$ for all $t \in T_1$ and $R^{c_1}_{|\cdot|}(t) > R^{c_2}_{|\cdot|}(t)$ for all $t \in T_2$.*

*Proof.* Without loss of generality, assume that $T_1 = \{t_1, \ldots, t_n\}$ and $T_2 = \{t'_1, \ldots, t'_n\}$ with $t_i < t'_i < t_{i+1}$ for $i \in \{1, \ldots, n\}$. We will construct $c_1, c_2$ such that the robustness curves $R^{c_1}_{|\cdot|}(\cdot), R^{c_2}_{|\cdot|}(\cdot)$ intersect at exactly the points $(t_i + t'_i)/2$ and $(t_i + t'_{i+1})/2$ on the interval $(t_1, t'_n]$. Let $d = t'_n$ and

$$P\left(-d - \frac{t_i + t'_{i+1}}{2}, 0\right) = P\left(d + \frac{t_i + t'_i}{2}, 1\right) = \frac{2}{4n+1}$$

and

$$P\left(-d - \frac{t_1}{2}, 0\right) = \frac{1}{4n+1} .$$

Let $c_1(x) = \mathbb{1}_{x \geq -d}$ and $c_2(x) = \mathbb{1}_{x \geq d}$. Both classifiers have perfect accuracy on $P$, meaning that $R^{c_i}_{|\cdot|}(0) = 0$. The closest point to the decision boundary of $c_1$ is $-d - \frac{t_1}{2}$ with weight $\frac{1}{4n+1}$, so $R^{c_1}_{|\cdot|}(\frac{t_1}{2}) = \frac{1}{4n+1}$. The second-closest point is $-d - \frac{t_1 + t'_2}{2}$ with weight $\frac{2}{4n+1}$, so $R^{c_1}_{|\cdot|}(\frac{t_1 + t'_2}{2}) = \frac{3}{4n+1}$, and so on. Meanwhile, the closest point to the decision boundary of $c_2$ is $d + \frac{t_1 + t'_1}{2}$ with weight $\frac{2}{4n+1}$, so $R^{c_2}_{|\cdot|}(\frac{t_1 + t'_1}{2}) = \frac{2}{4n+1}$, the second-closest point is $d\frac{t_2 + t'_2}{2}$ with weight $\frac{2}{4n+1}$, so $R^{c_2}_{|\cdot|}(\frac{t_2 + t'_2}{2}) = \frac{4}{4n+1}$, and so on.  □

**Example 1.** *To see that robustness curve intersections do not transfer between different $\ell_p$ norms, consider the example in Figure 6. The blue and orange linear classifiers both perfectly separate the displayed data. The $\ell_\infty$ robustness curves of the classifiers do not intersect, meaning that the robust error of the blue classifier is always better than that of the orange classifier. In $\ell_2$ distance, the robustness curves intersect, so that there is a range of perturbation sizes where the orange classifier has better robust error than the blue classifier.*

# B  ROBUSTNESS CURVE DEPENDENCE OF SHAPE ON DISTANCE FUNCTION

**Theorem 2.** *Let $f(x) = \mathrm{sgn}(w^T x + b)$ be a linear classifier. Then the* shape *of the robustness curve for $f$ regarding an $\ell_p$ norm-induced distance does not depend on the choice of $p$. It holds that*

$$R^f_{\ell_{p_1}}(\varepsilon) = R^f_{\ell_{p_2}}(c \cdot \varepsilon) \quad \forall \varepsilon \text{ for } c = \frac{\|w\|_{q_1}}{\|w\|_{q_2}}, q_i = \frac{p_i}{p_i - 1}. \tag{1}$$

**Lemma 1.** *Let $x \in \mathbb{R}^m$ with $w^T x + b \neq 0$. Let $p \in [1, \infty]$ and $q$ such that $\frac{1}{p} + \frac{1}{q} = 1$, where we take $\frac{1}{\infty} = 0$. Then*

$$\min\{\|\delta\|_p : \mathrm{sgn}(w^T(x + \delta) + b) \neq \mathrm{sgn}(w^T x + b)\} = \frac{|w^T + b|}{\|w\|_q} \tag{2}$$

*and the minimum is attained by*

$$\delta = \begin{cases} \frac{-w^T x - b}{\|w\|_\infty} \mathrm{sgn}(w_j) e_j, j = \mathrm{argmax}_i |w_i| & p = 1 \\ \frac{-w^T x - b}{\|w\|_q^q} (\mathrm{sgn}(w_i)|w_i|^{\frac{1}{p-1}})_{i=1}^d & p \in (1, \infty]. \end{cases} \tag{3}$$

*where $x^{\frac{1}{\infty - 1}} = x^0 = 1$ and $e_j$ is the $j$-th unit vector.*

*Proof of Theorem 2.* By Hölder's inequality, for any $\delta$,

$$\sum_{i=1}^m |w_i \delta_i| \leqslant \|\delta\|_p \|w\|_q. \tag{4}$$

For $\delta$ such that $\mathrm{sgn}(w^T(x + \delta) + b) \neq \mathrm{sgn}(w^T x + b)$ it follows that

$$\|\delta\|_p \geqslant \frac{\sum_{i=1}^m |w_i \delta_i|}{\|w\|_q} \geqslant \frac{|\sum_{i=1}^m w_i \delta_i|}{\|w\|_q} \geqslant \frac{|w^T x + b|}{\|w\|^q}. \tag{5}$$

Using the identity $q = \frac{p}{p-1}$, it is easy to check that for every $p \in [1, \infty]$, with $\delta$ as defined in Equation (3),

1.  $w^T \delta = -w^T x - b$, so that $w^T(x + \delta) + b = 0$, and

2.  $\|\delta\|_p = \frac{|w^T x + b|}{\|w\|_q}$.

Item 1 shows that $\delta$ is a feasible point, while Item 2 in combination with Equation (5) shows that $\|\delta\|_p$ is minimal. $\qquad\square$

Using Lemma 1, we are ready to prove Theorem 2.

*Proof.* By definition,

$$R^f_{\ell_{p_1}}(\varepsilon) = P(\underbrace{\{(x, y) \text{ s.t. } \exists \delta : \|\delta\|_{p_1} \leqslant \varepsilon \wedge f(x + \delta) \neq y\}}_{\mathcal{R}_{p_1}(\varepsilon)}). \tag{6}$$

We can split $\mathcal{R}_{p_1}(\varepsilon)$ into the disjoint sets

$$\underbrace{\{(x, y) : f(x) \neq y\}}_{=M} \tag{7}$$

$$\dot\cup \tag{8}$$

$$\underbrace{\{(x, y) \text{ s.t. } \exists \delta : \|\delta\|_{p_1} \leqslant \varepsilon \wedge y = f(x) \neq f(x + \delta)\}}_{=B_{p_1}(\varepsilon)}. \tag{9}$$

Choose $q_1, q_2$ such that $\frac{1}{p_i} + \frac{1}{q_i} = 1$. By Lemma 1, and using that $f(x) = \text{sgn}(w^T x + b)$,

$$B_{p_1}(\varepsilon) = \{(x, y) : \text{sgn}(w^T x + b) = y \wedge \frac{|w^T x + b|}{\|w\|_{q_1}} \leqslant \varepsilon\} \tag{10}$$

$$= \{(x, y) : \text{sgn}(w^T x + b) = y \wedge \frac{|w^T x + b|}{\|w\|_{q_2}} \leqslant \frac{\|w\|_{q_1}}{\|w\|_{q_2}} \varepsilon\}) \tag{11}$$

$$= B_{p_2}\left(\frac{\|w\|_{q_1}}{\|w\|_{q_2}} \varepsilon\right). \tag{12}$$

This shows that

$$R^f_{\ell_{p_1}}(\varepsilon) = P(M) + P(B_{p_1}(\varepsilon)) \tag{13}$$

$$= P(M) + P\left(B_{p_2}\left(\frac{\|w\|_{q_1}}{\|w\|_{q_2}} \varepsilon\right)\right) \tag{14}$$

$$= R^f_{\ell_{p_2}}\left(\frac{\|w\|_{q_1}}{\|w\|_{q_2}} \varepsilon\right). \tag{15}$$

$\square$

## C   EXPERIMENTAL DETAILS

### C.1   MODEL TRAINING

We use the same model architecture as Croce, Andriushchenko, and Hein (2019) and Wong and Kolter (2018). Unless explicitly stated otherwise, the trained models are taken from Croce, Andriushchenko, and Hein (2019). The exact architecture of the model is: Convolutional layer (number of filters: 16, size: 4x4, stride: 2), ReLu activation function, convolutional layer (number of filters: 32, size: 4x4, stride: 2), ReLu activation function, fully connected layer (number of units: 100), ReLu activation function, output layer (number of units depends on the number of classes). All models are trained with Adam Optimizer (Kingma and Ba 2014) for 100 epochs, with batch size 128 and a default learning rate of 0.001. More information on the training can be found in the experimental details section of the appendix of Croce, Andriushchenko, and Hein (2019). The trained models are those made publicly available by Croce, Andriushchenko, and Hein (2019)[5] and Croce and Hein (2020)[6].

### C.2   APPROXIMATED ROBUSTNESS CURVES

We use state-of-the-art adversarial attacks to approximate the true minimal distances of input data-points to the decision boundary of a classifier for our adversarial robustness curves (see Definition 1). We base our selection of attacks on the recommendations of Carlini, Athalye, et al. (2019). Specifically, we use the following attacks: For $\ell_2$ robustness curves we use the $\ell_2$-attack proposed by Carlini and Wagner (2017) and for $\ell_\infty$ robustness curves we use PGD (Madry et al. 2018). For both attacks, we use the implementations of Foolbox (Version 2.4) (Rauber et al. 2017). For the $\ell_\infty$ attack, the implementation of Foolbox automatically performs a hyperparameter search over different epsilon and uses the smallest resulting adversarial perturbation. For the rest of the hyperparameters, we use the standard values of the Foolbox implementation. For the $\ell_2$ attack, we increase the number of binary search steps that are used to find the optimal tradeoff-constant between distance and confidence from 5 to 10, which we found empirically to improve the results. For the rest of the hyperparameters, we again use the standard values of the Foolbox implementation.

### C.3   COMPUTATIONAL ARCHITECTURE

We executed all programs on an architecture with 2 x Intel Xeon(R) CPU E5-2640 v4 @ 2.4 GHz, 2 x Nvidia GeForce GTX 1080 TI 12G and 128 GB RAM.

---

[5] The models trained with ST, KW, AT and MMR + AT are avaible at
www.github.com/max-andr/provable-robustness-max-linear-regions.

[6] The models trained with MMR-UNIV are available at www.github.com/fra31/mmr-universal.

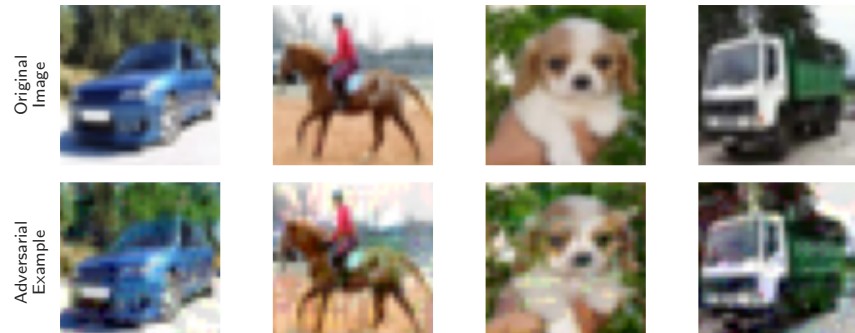

Figure 7: Visualization of four images from `CIFAR-10` (top row), together with adversarial examples (bottom row), calculated with PGD (Madry et al. 2018) for a model trained with `MMR + AT`, Threat Model: $\ell_\infty(\varepsilon = 2/255)$. The resulting perturbation sizes of the adversarial examples are (from left to right) 17/255, 18/255, 18/255, 18/255. Even for perturbation sizes far greater than popular choices of point-wise measures, adversarial examples can be very hard to detect for humans.

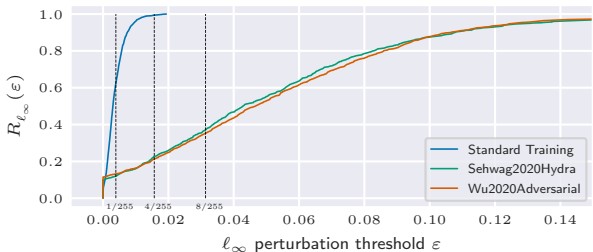

Figure 8: $\ell_\infty$ robustness curves for two state-of-the-art robust models with a large architecture (WideResNet-28-10). The labels indicate the training method (Sehwag2020Hydra: (Sehwag et al. 2020), Wu20Adversarial: (Wu et al. 2020)). The trained models are taken from Croce, Andriushchenko, Sehwag, et al. (2020). The models are trained on the full training set of `CIFAR-10`, and robustness curves are based on a sample of 1000 points from the test set.

## D    VISUALIZATION OF ADVERSARIAL EXAMPLES

As we pointed out in Section 1, adversarial robustness of classifiers trained on `CIFAR-10` is usually evaluated at a perturbation threshold $\varepsilon \in \{2/255, 4/255, 8/255\}$ for the $\ell_\infty$ norm. Robustness curves allow us to investigate robustness of classifiers for perturbation thresholds beyond those which are used in the literature. It should not be necessary for the model to be invariant under large perturbations, if these perturbations are clearly perceptible or change the "correct" classification of the input. However, the thresholds that models are currently optimized for are small enough that even larger perturbations may not be perceptible. Figure 7 shows four images of `CIFAR-10` (top row), together with adversarial examples (bottom row). With perturbation sizes $\varepsilon \in \{17/255, 18/255\}$, the perturbations are more than two times larger than the biggest perturbation threshold used in the literature, and still almost imperceptible for untrained humans.

## E    ROBUSTNESS CURVES FOR LARGER MODELS

In Section 3, we demonstrate the usefulness of robustness curves on a small convolutional network architecture used by Croce, Andriushchenko, and Hein (2019). The choice of a small architecture allows us to compute robustness curves for a large number of different defensive strategies with limited computational resources. Figure 8 shows approximate robustness curves for two state-of-the-art robust models with a large network architecture (WideResNet-28-10), computed for a sample of

1000 data points from `CIFAR-10`. Due to the small number of points used, the approximation may be rough, so the following observations should be taken with a grain of salt.

1. Both robust models are indeed much more robust than the model obtained by standard training even for perturbation thresholds that are significantly larger than the threshold of $8/255$ that the models are optimized for. This observation may help decide whether it is worthwhile to stop using a conventionally trained model, sacrificing accuracy for robustness.

2. Wu et al. (2020) has slightly worse accuracy than Sehwag et al. (2020) roughly up to perturbation size $1/255$. This is a trade-off for better accuracy between perturbation sizes $4/255$ and $0.1$. From perturbation size $0.1$ onward, Sehwag et al. (2020) appears to have slightly better accuracy than Wu et al. (2020). This observation may help decide which of the two robust models is preferable, based on the robustness requirements of a concrete application.

3. The gap between the performance of Wu et al. (2020) and Sehwag et al. (2020) is even wider at perturbation size $0.04$ than $8/255$, but overall, the robustness curves of the robust models are quite similar. This observation may help decide whether it is worthwhile to switch from one model to the other, if one of the models is already in use or preferable for other reasons.

