# OpenReview forum: "How to compare adversarial robustness of classifiers from a global perspective"
_ICLR.cc/2021/Conference — Reject_

### Official Review · AnonReviewer4 · 2020-10-27
**A good criterion for evaluating robustness of classifiers**

**Rating:** 6
**Confidence:** 3

**Review:**

Summary:
The paper showed that point-wise measures fail to capture important properties that are essential to compare the robustness of different classifiers. The authors introduced the recently proposed robustness curves to provide a global perspective including the scale as a way to distinguish small and large perturbations.

Pros:
(1) How to compare the robustness is a very important question for current machine learning models. The author introduced a better criterion for this important question.
(2) The paper is well written and easy to read.  The experiments and its discussion are strong proofs to support that the robustness curve should be the better criterion.
(3) The authors released code to reproduce all the experiments for the current popular frameworks. It will be very helpful for the researchers to the advantages of this curve over point-wise measures.

Cons:
Overall, this paper is impressive.
The only concern the reviewer has is the contribution compared to the previous work who proposed the robustness curve (C. Gopfert et al. 2020).  it seems this paper's contribution is highly based on  the proposal of robustness curve, and providing more explanations and discussions. (But this will not affect the importance of this paper)

---

> ### Author Response · Authors · 2020-11-15
> **Reply to AnonReviewer4**
>
> Thank you for your comments.
>
>
> > Cons: Overall, this paper is impressive. The only concern the reviewer has is the contribution compared to the previous work who proposed the robustness curve (C. Gopfert et al. 2020). it seems this paper's contribution is highly based on the proposal of robustness curve, and providing more explanations and discussions. (But this will not affect the importance of this paper)
>
> In the paper you mention ([C. Gopfert et al. 2020][4.1]) robustness curves are indeed introduced -- however, only their basic nature is explored therein. In our work, we provide several important contributions:
>
> 1. We show that, theoretically, robustness curves may intersect with arbitrarily many intersections, even for (but not limited to) linear classifiers (which seem relatively benign in [4.1]). See Figure 1 and Appendix A.
> 2. We show that intersections may appear or disappear while shifting between norms, even for linear classifiers. See Figure 6.
> 3. We demonstrate that such intersections occur in practice. See Figure 2.
> 4. We provide a ready-to-use implementation to calculate robustness curves for deep networks.
> 5. We investigate the transfer of robustness across distance functions for deep networks. See Section 3.2.3.
> 6. We establish a connection between the properties of datasets and robustness thresholds. See Section 3.3 and Table 2.
>
> Compared to [4.1], we show *how* to use robustness curves and explore the phenomena that become apparent by doing so. We hope that you agree and consider increasing your rating of our submission.
>
>
> [4.1]: https://arxiv.org/abs/1908.00096

---

### Official Review · AnonReviewer3 · 2020-10-28
**Marginally below acceptance threshold**

**Rating:** 5
**Confidence:** 4

**Review:**

This paper presents a theoretical scenario where point-wise measure of adversarial robustness falls short in comparing model robustness, then conduct experiments to show that robustness curve is a more meaningful evaluation metric from a global perspective.


Pros:


+ The motivation is well explained. I mainly agree with the authors on the argument that point-wise measurement of robustness may be insufficient in explaining model robustness. Computing and visualizing robustness curves seems to be more meaningful and rigorous from a security perspective.

+ Relating the choice of the perturbation strength to the underlying property of the data distribution is useful. The inter-class distances demonstrated in Table could potentially be used as a reference on determining the right scale of perturbation strength.


Cons:


- The robustness results presented in Table 1 seems far below the state-of-the-art robustness. For instance, in the last row (\epsilon=8/255), the robust test error of AT is 0.92, which is much higher than the reported statistics in (Madry et al., 2018). The author uses a very small 4-layer convolutional neural network for CIFAR-10 experiments, whereas the state-of-the-art robustness results are achieved using a much larger network, such as a ResNet architecture or a WideResNet architecture (refer to [1] for the current best robustness results on CIFAR-10). Thus, I recommend authors to rerun these experiments using a larger network.

- Similar architecture is used for the robustness curves in Figures 3 and 4. This suboptimal choice of network architecture makes the argument 'it contains multiple intersections in the robustness curve' unconvincing.


Other Questions or Comments:

1. Most of the existing defenses against adversarial examples are typically trained using a specifically-chosen perturbation strength. If adopting robustness curves (or global robustness) as the evaluation criteria instead of point-wise robustness, how will this affect the existing adversarial training procedure?

2. What does the distance statistics presented in Table 2 suggest for the typical choice of perturbation strength used in existing literature?

3. The global robustness considered in this paper is robustness for varying perturbation strength. Is there a way to define the perturbation strength for different input locations based on your computed inter-class statistics?

4. The bibliography style of the reference is not standard. Check if you are using the correct file.


[1] Reliable Evaluation of Adversarial Robustness with an Ensemble of Diverse Parameter-free Attacks, Francesco Croce and Matthias Hein, ICML 2020

---

> ### Author Response · Authors · 2020-11-15
> **Reply to AnonReviewer3**
>
> Thank you for your comments.
>
> > The robustness results presented in Table 1 seems far below the state-of-the-art robustness. For instance, in the last row (\epsilon=8/255), the robust test error of AT is 0.92, which is much higher than the reported statistics in (Madry et al., 2018). The author uses a very small 4-layer convolutional neural network for CIFAR-10 experiments, whereas the state-of-the-art robustness results are achieved using a much larger network, such as a ResNet architecture or a WideResNet architecture (refer to [1] for the current best robustness results on CIFAR-10). Thus, I recommend authors to rerun these experiments using a larger network.
> > Similar architecture is used for the robustness curves in Figures 3 and 4. This suboptimal choice of network architecture makes the argument 'it contains multiple intersections in the robustness curve' unconvincing.
>
> Our aim is to demonstrate the usefulness and importance of robustness curves with regards to the comparison between different defenses, so we calculate them for models used in specific recent works (see [3.1,3.2,3.3,3.4]) and for a number of defensive strategies. This implies a considerable amount of required compute, which we unfortunately lack access to. As such, we opt for simple architectures. Note that, as long as we compare the different strategies on one constant architecture, the comparison is still valid. We do, however, share your interest in robustness curves for better-performing deep networks, so we have chosen two models from [3.5] and performed the required calculations for a small sample of data points; we include the result in our Appendix (see Appendix E).
>
> [3.1]: Towards Deep Learning Models Resistant to Adversarial Attacks, Aleksander Madry, Aleksandar Makelov, Ludwig Schmidt, Dimitris Tsipras, and Adrian Vladu, ICLR 2018
>
> [3.2]: Provable Defenses against Adversarial Examples via the Convex Outer Adversarial Polytope, Eric Wong and Zico Kolter, ICML 2018
>
> [3.3]: Provable Robustness of ReLU networks via Maximization of Linear Regions, Francesco Croce, Maksym Andriushchenko, and Matthias Hein, AISTATS 2019
>
> [3.4]: Provable robustness against all adversarial l_p-perturbations for p >= 1, Francesco Croce and Matthias Hein, ICLR 2020
>
> [3.5]: https://robustbench.github.io/
>
> > Most of the existing defenses against adversarial examples are typically trained using a specifically-chosen perturbation strength. If adopting robustness curves (or global robustness) as the evaluation criteria instead of point-wise robustness, how will this affect the existing adversarial training procedure?
>
> Adopting a new, broader perspective on the evaluation of robustness will surely require adapting defensive strategies. One approach may be to iteratively robustify against larger and larger perturbations, but we cannot at this moment anticipate a concrete algorithm that will perform best according to robustness curves.
>
>
> > What does the distance statistics presented in Table 2 suggest for the typical choice of perturbation strength used in existing literature?
>
> For the most part, perturbation strengths in existing literature tend to be extremely small. Our results lead us to believe that larger choices would allow stronger results. We would have loved to provide a simple mapping from Table 2 to “ideal threshold”, but the issue is surely more involved than that. However, we can clearly see how different these common datasets are with regards to scale. As an example, we cannot expect a finding solely on MNIST to transfer to other datasets.
>
> > The global robustness considered in this paper is robustness for varying perturbation strength. Is there a way to define the perturbation strength for different input locations based on your computed inter-class statistics?
>
> We have not yet looked into regional effects on robustness thresholds, but we believe that your question points into an important direction for future work.
>
>
> > The bibliography style of the reference is not standard. Check if you are using the correct file.
>
> Thank you for pointing this out. We have adjusted the References accordingly.
>
>
> We hope that we addressed your comments to your satisfaction and would ask you to consider increasing your rating of our submission.

---

### Official Review · AnonReviewer1 · 2020-10-28
**Clear paper but importance of the problem not obvious.**

**Rating:** 5
**Confidence:** 4

**Review:**

Summary:
The authors advocate for the use of Robustness curves, plotting the adversarial accuracy as a function of the size of the neighbourhood region of allowed perturbation. The problem that they identify is that if you only evaluate adversarial accuracy at some numbers of threshold, you might conclude that some models (and the method that was used to train them) are more robust than others while it would be incorrect at other thresholds.

General comment:
The paper clearly describes the problem it deals with and reads easily. On the other hand, I am not entirely convinced by the importance of the problem. If you have a particular specification that you care about  ("Robustness against l_inf attack for eps=0.1"), you can just verify that. On the other hand, if your goal is "I want my network to be robust", then it's not properly defined, so of course it's hard to evaluate. Robustness curves will help there but there is still the problem that you might want to be robust to L_infinity, L_1, L_2, brightness difference, Wasserstein difference, changes of small patches... and then the robustness curves will not help you (unless of course you compute one for each difference). At the same time, they are quite a bit more costly to compute that simple point measures. While I agree that you are going to get more information if you compute a full robustness curve than if you sample it at a bunch of points, I'm not convinced that it is worth the effort.

One thing that I would recommend the authors is to make clearer the distinction between robustness curves as they described them (based on finding the closest adversarial example) vs. plotting on a robustness curve the pointwise measures and interpolating through them. This would be a much cheaper solution (and is essentially what reporting experimental results for a few chosen eps achieves). For example, the results the authors give in their Table 1. based on point wise measures wessentially achieves what the authors want to show: no defense strictly dominate the others.

Specific Comments:
- The toy dataset example presented in Figure 1. is great and provides a great explanation of the problem that the authors identify. The constructive proof in Appendix A. is also quite interesting and really drives the point that the authors want to make.

- The authors argue that robustness curves allow to compare "global robustness properties and their dependence on a given classifier, distribution and distance function". In practice, does it really give insights global robustness property? If I look at the L_infinity robustness curves, it does not tell me much about the robustness to L_2 perturbations.

- I don't understand how the robustness curves are generated for the L-infinity case? If PGD is used to find adversarial examples, it's not likely to be the "closest" adversarial examples that is going to be found, in all likelihood it's going to be one that matches the epsilon given as input to the PGD attack (due to the projection)? There is nothing that is even encouraging the sample to be close to the input beyond the constraints used for projection.
Even for the L2 distance, there is still the usual problem that, although the CW attack encourages to find the closest sample, there is no guarantee that it will, and the effectiveness it will have at doing so might depend from model to model. As a result, it's hard to decouple the robustness curve from the attack that it used internally.

Minor Notes & Typos:
- To improve the look of the paper, it should be possible to include manual linebreaks in the title so that it's not broken in every line.
- The authors talk in the introduction about "recently proposed robustness curves" and cite a paper from 2020 for them, but it seems like those curves were already in use before that. "On the effectiveness of interval bound propagation for training verifiably robust models", Gowal et al. had some in 2018.; "Provable Defenses against Adversarial Examples via the Convex Outer Adversarial Polytope" had some (transposed) in 2017.
- At the end of the introduction, the author say: "It is our belief that the continued use of single perturbation thresholds in the adversarial robustness literature is due to a lack of awareness of the shortcomings of these measures".  This seems overtly harsh. You could make the same point about training algorithms and say that authors only reporting on only a few datasets due it just out of lack of awareness of the fact that the relative performance of different algorithms will vary depending on the dataset. Given that computing robustness curves needs computing the closest adversary to a point, this is much more expensive so maybe computational cost might be the differentiating factor rather than "lack of awareness"?

---

> ### Author Response · Authors · 2020-11-15
> **Reply to AnonReviewer1 2/2**
>
> > The authors argue that robustness curves allow to compare "global robustness properties and their dependence on a given classifier, distribution and distance function". In practice, does it really give insights global robustness property? If I look at the L_infinity robustness curves, it does not tell me much about the robustness to L_2 perturbations.
>
>
> Yes, the problem is even larger. We mean “global” in a certain sense: the benefit of robustness curves is that they reveal the global robustness for a fixed classifier, distribution and distance function. This makes comparisons such as in Figures 2, 3 and 4 possible. We have updated Section 2 accordingly to improve clarity. Indeed, the robustness curve for one distance function contains little information about that for another distance function (an exception being linear models, which are somewhat benign), and none about that for other classifiers or distributions.
>
>
> > I don't understand how the robustness curves are generated for the L-infinity case? If PGD is used to find adversarial examples, it's not likely to be the "closest" adversarial examples that is going to be found, in all likelihood it's going to be one that matches the epsilon given as input to the PGD attack (due to the projection)? There is nothing that is even encouraging the sample to be close to the input beyond the constraints used for projection. Even for the L2 distance, there is still the usual problem that, although the CW attack encourages to find the closest sample, there is no guarantee that it will, and the effectiveness it will have at doing so might depend from model to model. As a result, it's hard to decouple the robustness curve from the attack that it used internally.
>
> The PDG implementation used for the l_infinity curves (Foolbox Version 2.4, ProjectedGradientDescentAttack) performs a hyperparameter search over different epsilon and uses the smallest resulting adversarial perturbation. We have clarified this in the corresponding section in the Appendix (see Appendix C.2). In general, it is true that the quality of the robustness curve approximation depends on the strength of the attack used. The same issue occurs for point-wise measures, where evaluation using adversarial attacks is the standard practice.
>
>
> > Minor Notes & Typos
>
> We have addressed your minor comments and typos and incorporated respective changes where possible. Thank you very much for raising these points, especially the last one! We surely want to prevent our wording from appearing overly harsh.
>
>
> We believe that your comments have helped us to raise the quality of our submission, and hope that you are willing to recommend acceptance.

---

> ### Author Response · Authors · 2020-11-15
> **Reply to AnonReviewer1 1/2**
>
> Thank you for the thoughtful and thorough comments!
>
>
> > The paper clearly describes the problem it deals with and reads easily. On the other hand, I am not entirely convinced by the importance of the problem. If you have a particular specification that you care about ("Robustness against l_inf attack for eps=0.1"), you can just verify that. On the other hand, if your goal is "I want my network to be robust", then it's not properly defined, so of course it's hard to evaluate. Robustness curves will help there but there is still the problem that you might want to be robust to L_infinity, L_1, L_2, brightness difference, Wasserstein difference, changes of small patches... and then the robustness curves will not help you (unless of course you compute one for each difference). At the same time, they are quite a bit more costly to compute that simple point measures. While I agree that you are going to get more information if you compute a full robustness curve than if you sample it at a bunch of points, I'm not convinced that it is worth the effort.
>
> We agree that adversarial robustness is not yet well-defined. Ultimately, an adversarially robust model should be robust against various (reasonable) perturbation magnitudes as well as various types of perturbations, such as the ones you pointed out. Most current work is focused on robustness with respect to a single such type, and we argue that the frequently used single-threshold evaluations are not enough to convincingly show that one model is truly superior to another. For robustness against multiple such types of perturbations, recent work uses worst-case robustness under the union of multiple point-wise threat models as an evaluation [2.1,2.2]. Robustness curves can be extended to multiple threat models accordingly by defining an equivalence of perturbation sizes between different perturbation types. Ultimately, unions of threat models will likely also not be sufficient to achieve “general” robustness, since semantic robustness is not well approximated by currently used distance functions [2.3,2.4]. Of course, we do not presume that our contribution could be a final answer to the general ill-defined mystery around adversarial robustness. Instead, we hope that we can encourage a shift of focus away from point-wise robustness towards a discussion of the question what kind of robustness could and should reasonably be expected. We have adapted our abstract to make this key message more clear.
>
> [2.1]: Adversarial Training and Robustness for Multiple Perturbations, Florian Tramèr, Dan Boneh, NeurIPS 2019
>
> [2.2]: Adversarial Robustness Against the Union of Multiple Perturbation Models, Pratyush Maini, Eric Wong, J. Zico Kolter, ICML 2020
>
> [2.3]: Adversarial attacks hidden in plain sight, Jan Philip Göpfert, Heiko Wersing, Barbara Hammer, IDA 2020
>
> [2.4]: Breaking certified defenses: semantic adversarial examples with spoofed robustness certificates, Amin Ghiasi, Ali Shafahi, Tom Goldstein, ICLR 2020
>
>
> > One thing that I would recommend the authors is to make clearer the distinction between robustness curves as they described them (based on finding the closest adversarial example) vs. plotting on a robustness curve the pointwise measures and interpolating through them. This would be a much cheaper solution (and is essentially what reporting experimental results for a few chosen eps achieves). For example, the results the authors give in their Table 1. based on point wise measures wessentially achieves what the authors want to show: no defense strictly dominate the others.
>
> In the limit, computing enough point-wise measures yields the robustness curve. However, since robustness curves may intersect at arbitrarily many points, sampling a small set of thresholds may give misleading results: sampling point-wise measures at points $p_1,...,p_n$ could give the impression that one model is strictly more robust than the other on the interval $[p_1, p_n]$ when in fact, there are intersections in each interval $[p_i, p_i+1]$. In practice, a large number of intersections is unlikely. It would be interesting to investigate how closely the curves need to be approximated in order to estimate the number of intersections, if any, and their location, with high certainty. We have added this to the discussion as potential future work.

---

### Official Review · AnonReviewer2 · 2020-10-29
**This paper provides some interesting insights.**

**Rating:** 6
**Confidence:** 3

**Review:**

This paper surveys various adversarial defense methods on their performance when the perturbation distortion epsilon is increased. The author argues that robustness for a specific epsilon may not be enough and suggests robustness curves as an alternative. I think in general this paper provides some interesting empirical studies. My detailed comments are as follows.
1. The overfitting of specific epsilon value is expected but interesting to see and I think this is one of the main reasons why a robustness curve is necessary. However, in this adversarial robustness community, the status quo is that researchers compare with each other on some specific datasets with some specific epsilon, for example, 0.3 for MNIST, 8/255 for CIFAR. I think one reason for choosing these values is that studying robustness under perturbation with larger distortion is kind of unnecessary because then the noise added is no longer imperceptible, which is at odds with adversarial examples' definition. I think the authors may need to provide more discussion on why studying robustness with epsilon>0.6 for MNIST is necessary, since we may misclassify those images as human.
2. The authors suggest a robustness curve as an evaluation metric. However, I haven't seen any works on improving the robustness for all epsilon values globally. One possibility is that there is a trade-off between small epsilon performance and large epsilon performance, similar to the trade-off between robustness and accuracy (epsilon=0 performance). My suggestion is that the authors may define an "area under the curve" value just like in ROC curves for better comparison.
3. (Minor) Please refrain from only using color to distinguish curves in figures as it may not be friendly to readers with color blindness.

---

> ### Author Response · Authors · 2020-11-15
> **Reply to AnonReviewer2**
>
> Thank you for your thoughtful comments.
>
>
> > The overfitting of specific epsilon value is expected but interesting to see and I think this is one of the main reasons why a robustness curve is necessary. However, in this adversarial robustness community, the status quo is that researchers compare with each other on some specific datasets with some specific epsilon, for example, 0.3 for MNIST, 8/255 for CIFAR. I think one reason for choosing these values is that studying robustness under perturbation with larger distortion is kind of unnecessary because then the noise added is no longer imperceptible, which is at odds with adversarial examples' definition. I think the authors may need to provide more discussion on why studying robustness with epsilon>0.6 for MNIST is necessary, since we may misclassify those images as human.
>
> We agree that it should not be necessary for the model to be invariant under large perturbations, if these perturbations are clearly perceptible or change the “correct” classification of the input. However, the thresholds that models are currently optimized for are so small that even perturbations above said thresholds may not be considerably more perceivable: the selected thresholds seem to depend more on what is achievable than what is desirable. We have added examples for this to the Appendix (see Appendix D). Note that it is still unclear what exactly constitutes a small perturbation or a large perturbation, even -- or especially -- with regards to human perception. This uncertainty is precisely why we believe it necessary to find a broader perspective and deeper understanding of robustness beyond those specific choices of perturbation sizes.
>
>
> > The authors suggest a robustness curve as an evaluation metric. However, I haven't seen any works on improving the robustness for all epsilon values globally. One possibility is that there is a trade-off between small epsilon performance and large epsilon performance, similar to the trade-off between robustness and accuracy (epsilon=0 performance). My suggestion is that the authors may define an "area under the curve" value just like in ROC curves for better comparison.
>
> Yes, absolutely. The example we give in Figure 1 hinges on that tradeoff between robustness against small perturbations vs robustness against large ones. This underlines the importance of the global perspective on robustness that we offer. In line with your suggestion, we searched for single indicative numbers to summarize the curves, to be able to more directly perform comparisons. However, because there can be any number of intersections between robustness curves, determining how far one can condense the information contained in them with regards to practical settings will require further research.
>
>
> > (Minor) Please refrain from only using color to distinguish curves in figures as it may not be friendly to readers with color blindness.
>
> Thank you for noticing this! We have switched all figures to a colorblind-friendly palette to remedy this oversight of ours.
>
>
> We hope that, in response to our reply, you consider increasing your rating of our submission.

---

### Decision · Program_Chairs · 2021-01-07
**Final Decision**

**Decision:**

Reject

**Comment:**

The authors study "robustness curves" which are plots of the robust error versus the radius used in the corresponding l_p-ball threat model.

Pro: I completely agree with the authors that the current evaluation purely based on evaluation for a single radius is insufficient
and one should report the complete curve.

Con: The authors are overclaiming that they have come up with robustness curves. Very early papers e.g. even in the adversarial
training paper of Madry there are plots of robust accuracy versus chosen threshold. Moreover, I agree with one of the reviewers that using PGD for the purpose of a robustness curve is inaccurate and in particular inefficient as several attacks for different radii have to be done. There have been several attacks developed which aim to find the adversarial sample with minimum norm and thus compute the robustness curve in one run.

The additional insights e.g. intersection of robustness curves are partially to be expected and I don't find them sufficient to move the paper over the bar for ICLR.  As these insights are additionally  only shown for relatively small models which seem far away from the state of the art, it is unclear if they generalize. However, I encourage to follow some of the reviewer's suggestions to improve the paper.